

# Minimum information about tolerogenic antigen-presenting cells (MITAP): a first step towards reproducibility and standardisation of cellular therapies

Phillip Lord[1,*], Rachel Spiering[2], Juan C. Aguillon[3], Amy E. Anderson[2], Silke Appel[4], Daniel Benitez-Ribas[5], Anja ten Brinke[6], Femke Broere[7], Nathalie Cools[8], Maria Cristina Cuturi[9], Julie Diboll[2], Edward K. Geissler[10], Nick Giannoukakis[11], Silvia Gregori[12], S. Marieke van Ham[6], Staci Lattimer[1], Lindsay Marshall[1], Rachel A. Harry[2], James A. Hutchinson[10], John D. Isaacs[2], Irma Joosten[13], Cees van Kooten[14], Ascension Lopez Diaz de Cerio[15], Tatjana Nikolic[16], Haluk Barbaros Oral[17], Ljiljana Sofronic-Milosavljevic[18], Thomas Ritter[19], Paloma Riquelme[10], Angus W. Thomson[20], Massimo Trucco[11], Marta Vives-Pi[21,22], Eva M. Martinez-Caceres[21,23] and Catharien M.U. Hilkens[2,*]

[1] School of Computing Science, Newcastle University, Newcastle upon Tyne, United Kingdom
[2] Institute of Cellular Medicine, Newcastle University, Newcastle upon Tyne, United Kingdom
[3] Instituto de Ciencias Biomédicas (ICBM), Universidad de Chile, Santiago, Chile
[4] Broegelmann Research Laboratory, Department of Clinical Science, University of Bergen, Bergen, Norway
[5] Department of Immunology, Hospital Clínic i Provincial and Centro de Investigación Biomédica en Red de Enfermedades Hepáticas y Digestivas (CIBERehd), Barcelona, Spain
[6] Department of Immunopathology, Sanquin Research, Amsterdam, The Netherlands
[7] Faculty of Veterinary Medicine, Department of Infectious Diseases and Immunology, Utrecht University, Utrecht, The Netherlands
[8] Laboratory of Experimental Hematology, Vaccine & Infectious Disease Institute (VAXINFECTIO), Faculty of Medicine and Health Sciences, University of Antwerp, Wilrijk, Belgium
[9] Center for Research in Transplantation and Immunology, ITUN, Inserm UMRS 1064, Nantes, France
[10] Department of Surgery, Section of Experimental Surgery, University Hospital Regensburg, Regensburg, Germany
[11] Institute of Cellular Therapeutics, Allegheny Health Network, Pittsburgh, PA, United States of America
[12] San Raffaele Telethon Institute for Gene Therapy (SR-Tiget), Division of Regenerative Medicine, Stem Cells and Gene Therapy, IRCCS San Raffaele Scientific Institute, Milan, Italy
[13] Department of Laboratory Medicine, Radboud University medical center, Nijmegen, The Netherlands
[14] Department of Nephrology, Leiden University Medical Centre, Leiden, The Netherlands
[15] Area of Cell Therapy, University Clinic of Navarra, Pamplona, Spain
[16] Department of Immunohematology and Blood Transfusion, Leiden University Medical Centre, Leiden, The Netherlands
[17] Department of Immunology, Faculty of Medicine, Uludag University, Bursa, Turkey
[18] Institute for the Application of Nuclear Energy INEP, University of Belgrade, Belgrade, Serbia
[19] Regenerative Medicine Institute (REMEDI), School of Medicine, College of Medicine, Nursing and Health Sciences, National University of Ireland, Galway, Ireland
[20] Thomas E. Starzl Transplantation Institute, University of Pittsburgh School of Medicine, Pittsburgh, PA, United States of America
[21] Immunology Division, Germans Trias i Pujol University Hospital and Health Sciences Research Institute, Badalona, Spain
[22] CIBER of Diabetes and Associated Metabolic Diseases (CIBERDEM), Instituto de Salud Carlos III (ISCIII), Madrid, Spain
[23] Department of Cell Biology, Physiology, Immunology, Universitat Autònoma, Barcelona
[*] These authors contributed equally to this work.

Corresponding authors
Phillip Lord,
phillip.lord@newcastle.ac.uk
Catharien M.U. Hilkens,
catharien.hilkens@newcastle.ac.uk

## ABSTRACT

Cellular therapies with tolerogenic antigen-presenting cells (tolAPC) show great promise for the treatment of autoimmune diseases and for the prevention of destructive immune responses after transplantation. The methodologies for generating tolAPC vary greatly between different laboratories, making it difficult to compare data from different studies; thus constituting a major hurdle for the development of standardised tolAPC therapeutic products. Here we describe an initiative by members of the tolAPC field to generate a minimum information model for tolAPC (MITAP), providing a reporting framework that will make differences and similarities between tolAPC products transparent. In this way, MITAP constitutes a first but important step towards the production of standardised and reproducible tolAPC for clinical application.

el, Antigen-presenting cells, Autoimmune disease, Transplantation, Tolerogenic dendritic cells, Regulatory macrophages, Reporting guidelines

## INTRODUCTION

Immunotherapy with whole living cells shows great promise for the treatment of a wide variety of complex diseases. A recent innovation in this field is the use of myeloid antigen-presenting cells (APC) with immunoregulatory function to treat autoimmune diseases, or to prevent destructive immune reactions after organ—or haematopoietic stem cell-transplantation (*Amodio & Gregori, 2012*; *Hutchinson, Riquelme & Geissler, 2012*; *Hilkens & Isaacs, 2013*; *Van Brussel et al., 2014*; *Creusot et al., 2014*; *Thomas, 2014*; *Morelli & Thomson, 2014*; *Ten Brinke et al., 2015*). Examples of these immunoregulatory APC include tolerogenic dendritic cells (tolDC) and regulatory macrophages (Mreg). The main advantage of these tolerogenic/regulatory APC (hereafter referred to as tolAPC) therapies over conventional immunosuppressive drugs is that these cells are uniquely equipped to specifically target harmful T-cell responses to auto- or allo-antigens. Thus, tolAPC act in an antigen-specific manner and do not cause general immunosuppression after exposure to antigen or in the allogeneic transplant setting—they are, therefore, less likely to affect protective immune responses to invading pathogens. More remarkably, as tolAPC induce or restore immune tolerance, and immune tolerance is a self-reinforcing state, the beneficial effects of tolAPC treatment are expected to continue beyond the lifespan of these therapeutic cells. Thus, tolAPC therapy holds the promise of a potentially curative treatment with low toxicity.

A variety of therapeutic tolAPC products have been developed over the last decade, and a number of these have recently been tested, or are undergoing testing in the clinic. Phase I trials with autologous tolAPC have been completed for type I diabetes (*Giannoukakis et al., 2011*), rheumatoid arthritis (*Benham et al., 2015*; *Bell et al., 2016*) and Crohn's disease (*Jauregui-Amezaga et al., 2015*), and further phase I trials for multiple sclerosis, allergic asthma and kidney transplantation are in progress (*Ten Brinke et al., 2015*). So far the results are highly encouraging from a safety standpoint, and further phase II clinical trials

to test the efficacy of tolAPC therapies are under development, with the first tolDC phase II trial in type I diabetes recruiting imminently.

An outstanding and important issue within the tolAPC field is the variation in the methods used for cell isolation and culture. Current protocols to produce tolDC or Mreg involve the isolation of peripheral blood CD14$^+$ monocytes followed by cell culture under specific conditions to generate the desired type of tolAPC. For example, monocytes can be differentiated into DC by culturing them in the presence of the cytokines IL-4 and GM-CSF, and stable tolerogenic function can be induced in these DC by treatment with immunosuppressive compounds (e.g., dexamethasone, IL-10) or genetic modification to silence immunostimulatory molecules. In addition to compound variety, the concentration used may vary, as well as the timing of administration to the tolAPC cultures. These almost endless variations in the methodology make comparison of different tolAPC products difficult, therefore bringing real uncertainty when comparing safety and efficacy results from different clinical trials testing different tolAPC products in different diseases/conditions.

## MINIMUM INFORMATION MODELS (MIMS)

Minimum Information Models (MIMs) are now being recognised as an important tool for the scientific community to use and interpret reported data, enabling comparison between data from different studies and facilitating experimental reproducibility. While it is simple for different laboratories to share and exchange their data, for this sharing to be meaningful, allowing reuse and repurposing, it must share some commonality of reporting, methodology or interpretation. This process has recently raised much interest, for example with the 5-star open data initiative (http://5stardata.info). A MIM is the first step in this process, providing a mechanism for the broadest level of commonality, ensuring that different laboratories report at least the key facts about their analysis in a clear and consistent manner, by providing a document and checklist describing the necessary information. They are simple and easy-to-use, so are supportive rather than burdensome for investigators. Examples of MIMs are MIAME (minimum information about a microarray experiment) (*Knudsen, Daston & Teratology, 2005*) and the more recent MIATA (minimal information about T-cell assays) (*Janetzki et al., 2009*; *Britten et al., 2011*), the latter now actively being endorsed by seven international scientific journals. As well as allowing sharing of data between individual laboratories, when widely used, they also provide an excellent basis for building large integrated resources of data that can be searched efficiently and comprehensively.

## SETTING UP MITAP: COMMUNITY BUILDING AND INITIAL ANALYSIS

Our goal was to provide a reporting framework that will make differences and similarities of tolAPC transparent. We therefore set out to create minimum reporting guidelines for the production process of tolAPC used in pre-clinical and/or clinical studies. We call this MITAP (Minimum Information about Tolerogenic Antigen-Presenting cells).

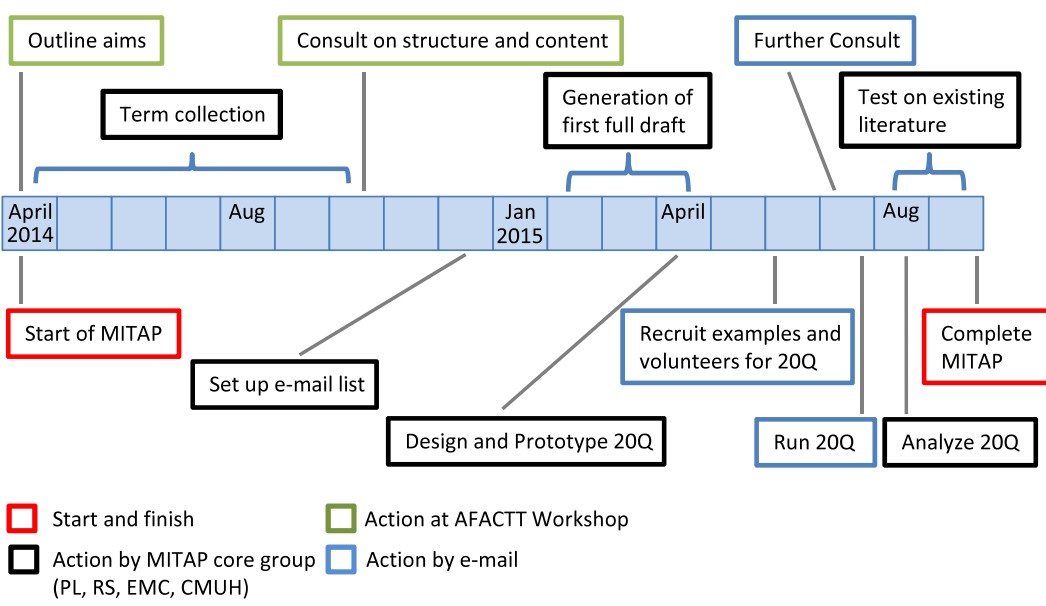

Figure 1 **MITAP time line: from start to finish.** 20Q: the "20 questions" experiment.

A critical part of building the MITAP document was community building; to enable data sharing and commonality, the document must be broadly reflective of the opinions and current practices of many laboratories within the discipline. For this, a community is needed to provide information, feedback and tests subjects. Helpfully, we recently established a consortium of investigators working in the field of tolerogenic cellular therapies, with the aim of jointly addressing issues related to the translation and clinical application of these new treatments. This consortium is called AFACTT (action to focus and accelerate cell-based tolerance-inducing therapies—http://www.afactt.eu). It convenes every 6 months at interactive workshops. This consortium provided an ideal vehicle for the process of developing MITAP, which we achieved through a series of "exercises" that introduced AFACTT members to the general notion of MIMs, as well as providing some initial data, which we used to scope the MITAP document. We used three main forms of exercise both of which aimed at gathering "terms"—basic vocabulary in use within the community. The first is called the "sticky-note" exercise which we performed at several AFACTT meetings: for this, each participant writes a term on a sticky-note; these are then collated and clustered on a wall by the whole group, identifying synonyms and related terms. Secondly, we gathered bibliographic databases from all willing AFACTT participants; abstracts were gathered from PubMed, and analysed computationally to find words that were over-represented in these abstracts compared to the bulk of PubMed. Finally, we built some bespoke software to crowd-source community feedback on these terms using a simple web interface.

The conception of MITAP took place in April 2014, at the first AFACTT meeting; the overall process of creating the MITAP document took 18 months (Fig. 1), which reflects the nature of the process i.e., the active involvement of a diverse group of investigators and the absolute need for incorporating feedback and suggestions from this group.

# OVERVIEW OF THE MITAP DOCUMENT

The MITAP document is divided into 4 sections that encapsulate the most important parts of the tolAPC production process in a chronological manner. The sections describe the cells at the start of the production process, during the differentiation/induction of tolerogenicity process, and at the end of the production process, prior to injection into the recipient or use in experimental assays. A final section is dedicated to aspects that are important for understanding the application and purpose of tolAPC. The sections are described in more detail below; for all the sections and subsections the document states clearly whether the investigator *must* (required), *should* (if available) or *may* (optional) provide the relevant information. It also advises on the use of existing taxonomies and databases to provide the information in a uniform manner, and it suggests the use of other MIMs where appropriate. The full MITAP document can be found on http://w3id.org/ontolink/mitap (see also 'Sustainability' below).

## Section 1. Cells before

This section describes the characteristics and state of the cells *before* they undergo any manipulation to become a tolAPC. There are five subparts to this section. First, it asks for essential information about the donor, including the species (surprisingly not always mentioned in research papers), strain if working with experimental animals, and any characteristics of the organism that are deemed to have a possible impact on the tolAPC production process (e.g., age or sex of the organism, or whether they are taking medication). Second, the tissue, organ or fluid from which the cells are used need to be described. In most published tolAPC studies the starting material is bone marrow or peripheral blood, but it is conceivable that for some applications investigators may want to isolate cells from specific organs (e.g., spleen). For human studies, peripheral blood products are the most often used source; these include freshly drawn blood by venepuncture, leukapheresis products and buffy coats (the latter can be purchased from Bloodbanks). Third, if investigators extract a subpopulation of cells from the initial cell source then information about the extraction method and the equipment should be provided. Fourth, the phenotype of the extracted cells should be described (e.g., morphology, expression of cell markers); in addition, the proportion of the cells that display a certain characteristic needs to be reported, to provide important information on the uniformity ('purity') of the cell population. In the last subpart of this section, details on the absolute cell number and cell viability should be provided.

## Section 2. Differentiation and induction of tolerogenicity

This section describes the protocol that has been used to differentiate and/or induce tolerogenicity in the cells described in Section 1. It comprises five parts. The first part describes the pre-culture conditions the cells are being kept in before the start of the cell culture process to generate tolAPC. This may include freezing and thawing of the cells. Second, the culture conditions of the cells should be provided, including the starting cell number, cell concentration, culture medium, culture container, and the culture environment (e.g., temperature, $CO_2$ levels). The third part deals with the differentiation/induction of tolerogenic function in the cells. It should be noted here that

differentiation and induction of tolerogenicity are not necessarily interchangeable; they can be seen as distinct processes. For example, tolAPC have been generated by differentiation of $CD14^+$ monocytes into immature or semi-mature DC; thus, without any active induction of pro-tolerogenic properties in the cells. In contrast, other protocols rely on the use of specific agents to induce stable tolerogenic function in DC during differentiation from precursor cells, and thus will use agents for both the differentiation and conferral of tolerogenicity of/in cells. In the fourth part investigators are asked to describe the antigen (if any) they use to load the tolAPC; this part is more likely to apply to investigators working in the field of autoimmunity, where tolAPC need to be targeted to certain autoantigen(s), and less so to the field of transplantation, which often (but not exclusively) uses donor-derived tolAPC already expressing the relevant allogeneic MHC/peptide complexes. Use of autologous APCs pulsed with donor antigen, e.g., in the form of donor cell lysate or exosomes (that accommodates the polymorphic MHC) is also possible. The final part of this section is about storage of the cells. If tolAPC are administered 'freshly,' the conditions under which the cells are being kept in between harvesting and injection into the recipient or use in experimental assays need to be described. On the other hand, if tolAPC are being frozen, the process of freezing and thawing needs to be described.

### Section 3. Cells after

This section describes the characteristics and state of the cells *after* the differentiation/induction of tolerogenicity process has taken place. Two of the parts (parts 1 and 3) cover the same type of information as in Section 1, i.e., details on the cell phenotype, number and viability. The second part of this section provides details on any characteristics of the cells that have been measured by means of a functional *in vitro* assay. This can include details on their migratory capacity, or their ability to induce T regulatory cells.

### Section 4. About the protocol

The final section of MITAP describes the general features of the protocol as a whole. The first part provides information about any external regulatory authorities that have approved the study, and whether the cells were produced under GLP or GMP guidelines. Second, the purpose of the produced tolAPC should be described, e.g., prevention of transplant rejection or restoring immune tolerance in an autoimmune disease. The relationship between the cells and the target organism should be described i.e., allogeneic, autologous, xenogeneic or syngeneic. Finally, the name and contact details of the corresponding author must be provided.

The MITAP document is accompanied by a handy checklist to assist investigators in ensuring that all the relevant detail is provided before submitting their manuscripts for publication.

## TESTING MITAP: THE "20 QUESTIONS" EXPERIMENT

The MITAP document was developed collaboratively by scientists in the AFACTT consortium and should, therefore, be usable by this community. However, we wished to test directly whether the document as it stood was comprehensible by individuals
from the discipline, and that data could be entered in a consistent manner. We therefore performed a "question-and-answer" experiment, a variation on the spoken parlour game "20 questions," popular in the 19th and 20th centuries. We collected a list of *terms*—that is short descriptive phrases that might be found in a description of a procedure to generate tolAPC, each of which would fit into one of the categories defined by the MITAP document. We then asked a number of scientists from the discipline, our *test subjects*, to place these terms into the relevant category from MITAP, which we call the *test task*. The basic premise of the experiment is simple: if the document is clear and comprehensible, then different test subjects should independently perform this task consistently.

The experiment was designed following a number of preconditions. First, the test task could not be too onerous for the test subjects, so that we could maximise completion rates; in particular, we felt that the task should take less than 30 min (including reading documentation). Our initial trials suggested that a subject could place two phrases for each category in MITAP in this time. Second, our terms should be representative of the discipline, rather than a single group, as far as was possible. Therefore, we asked members of the AFACTT group to contribute these terms, limited to two for each section in MITAP; for the experiment, we selected a subset of these terms randomly, with minor alterations for length. Third, our test subjects should also be representative of the discipline. Therefore, we asked those who contributed terms to also nominate a test subject. Finally, the test subjects were people who had not directly contributed to MITAP, nor seen the document, nor were involved in contributing terms.

For the test itself, test subjects were presented with a set of terms and for each asked to assess into which category it should be placed. Test subjects were given no guidance on how to categorise terms except for the MITAP document. In fact, they were given a slightly reduced MITAP document; the non-normative (i.e., explanatory and not a strict part of the standard) examples were removed for the purpose of this experiment. Subjects were also asked for some basic metadata (level of experience, discipline and so forth). The test was conducted using an online survey system (Google forms). Strictly, this means that we could not enforce lack of collaboration; we considered this compromise to be appropriate, as it increased the number of test subjects and their spread across the discipline.

## "20 QUESTIONS" RESULTS

The question and answer experiment was conducted over a two-week period in July 2015. We invited 24 test subjects to take part and received results from 23, representing a 96% completion rate. Each subject categorized 44 terms from which example terms are "xenogeneic," "*Rattus norvegicus*" or "immunomagnetic separation."

For the test task, subjects could ascribe each term to any number of MITAP categories. A number of categories in MITAP are repetitive (cell numbers before and after the differentiation/induction of tolerogenicity process, for example). These categories were normalized for most of the results presented below. Answers from test subjects were compared against our "Gold Standard"—the answers defined by primary authors of the MITAP document. As subjects could give multiple answers, we assessed answers as either being exactly the same as the gold standard, or containing at least the gold standard (Fig. 2).

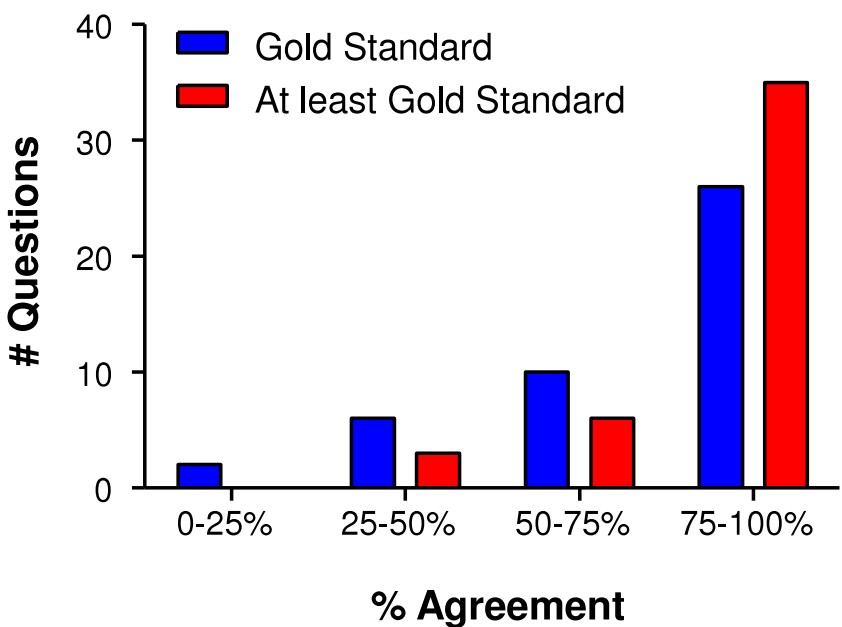

**Figure 2** **Summary of the results of the "20 Questions" Experiment.** The answers of the test subjects were divided over four quadrants: 0–25%, 25–50%, 50–75% and 75–100%, where e.g., 0–25% means that the number of questions that were only answered correctly by 0-25% of the test subjects and 75–100% meaning that these questions were answered correctly by 75–100% of the participants. A total of 44 questions were presented to test subjects. In blue, we show only answers with exactly the Gold Standard as correct.

Broadly, the results showed a very high-level of test subject agreement, averaging at around 73% for Gold Standard only and 84% for at least Gold Standard. Terms that were put in the right category by all test subjects included 'immunomagnetic separation' (Section 1. Cell separation process) and 'Spanish regulatory agency (AEMPS)' (Section 4. Regulatory Authority). Terms that test subjects were struggling with (0–50% agreement) included "cytomegalovirus (CMV)" for which the Gold Standard was antigen (see Section 2. Differentiation and Induction of Tolerogenicity), but that could also correctly be put under 'characteristics of the organism' (Section 1). Another example of a term often placed under the wrong subheading was: "mice were obtained from the Jackson Laboratory," which was often placed under 'species and strain' (Section 1), but for which the Gold Standard was 'characteristics of the organism' (Section 1). For these terms, where the document was shown to lack in clarity, we adjusted the MITAP document accordingly; for example, we now state explicitly under 'characteristics of the organism' that authors should indicate the source of purchase.

## PREVALENCE OF MITAP DATA IN EXTANT PAPERS

The purpose of the MITAP document is to ensure that authors provide sufficient basic information about their tolAPC production protocol. An implicit assumption is that

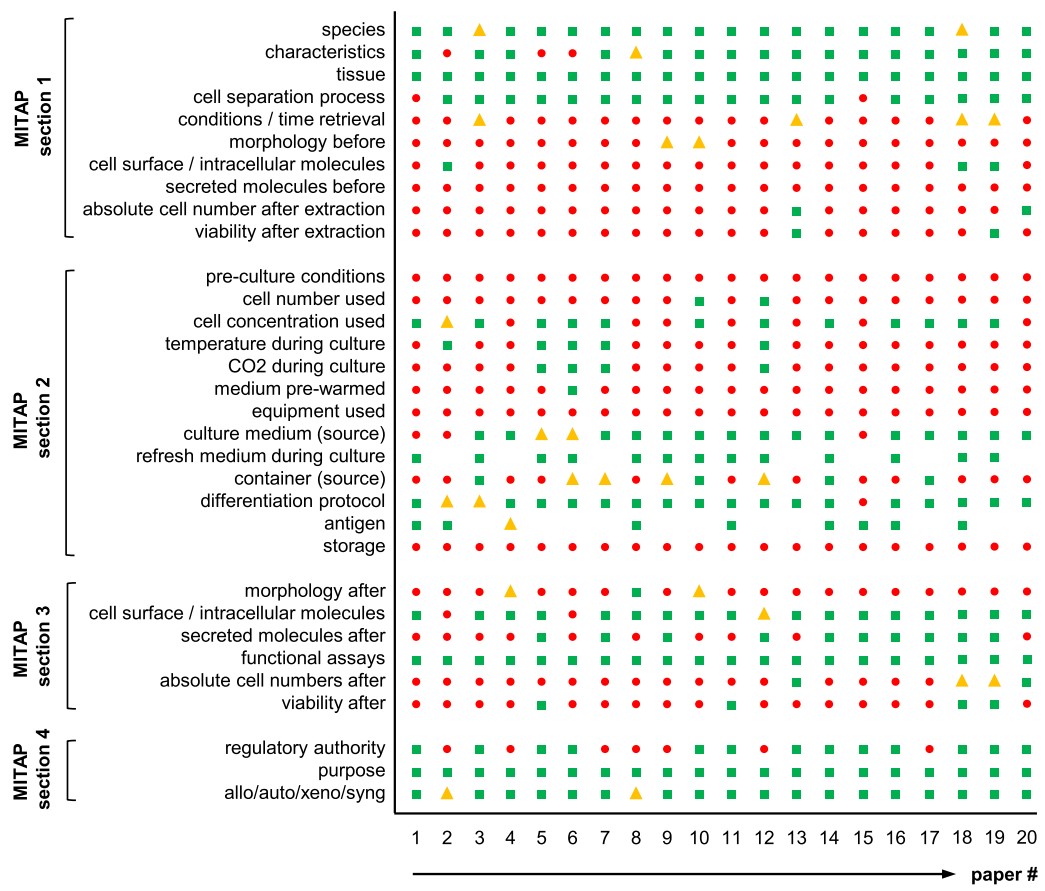

**Figure 3  Agreement of published tolDC papers with the MITAP document.** Graph showing the results of a total of twenty tolAPC papers. Green square: category included in the publication; Yellow triangle: category included in the publication, but some details missing; Red circle: category not included in the publication; -gap- information not present because not relevant for the publication.

currently some or all of this information is not being routinely described. To test this assumption, we reviewed a number of papers about tolAPC, and for each, we determined whether the data required by the MITAP document was present or not.

In detail, forty-two tolAPC papers were selected (predominantly from members of AFACTT or from researchers wellknown in the field), randomized and the first twenty were read in detail. For each section of MITAP, we determined whether the information required was either: directly stated in the paper or reference (Fig. 3: green squares), whether it was possible to infer from other information given in the paper (Fig. 3: yellow triangles), whether all details were present or only partly (Fig. 3: yellow triangles) or whether the information was not present at all (Fig. 3: red circles). For example, Section 1-ai of MITAP describes the species of the experimental organism. A paper with the phrase "human" or "*Homo sapiens*" would fall into the first category (*included in the publication*) one referring to "patients" would fall into the second category (*included but some details missing*). Many papers do not describe their experimental methodology, but instead refer to another paper ("as described previously"); in this case, we checked the paper up to two references away

and if found the information was considered as 'present' (Fig. 3: green squares), if not it was considered as 'not present' (Fig. 3: red circles). This work was performed by one of us (RS) who is a post-doctoral scientist with 6 years experience in the discipline.

Results are shown in Fig. 3. This shows that no papers have all the information required by MITAP, and that most have substantial gaps. We can also see that while there is a relatively high degree of conformity between different papers: e.g., species and growth media are commonly reported; cell number, viability and culture container are rarely documented.

## SUSTAINABILITY

One significant issue with resources such as MITAP is their sustainability. Previous research has shown that uniform resource identifiers (URIs; i.e., URLs or Web addresses) given in papers have short half-lives (around a 25% loss three years after publication (*Wren, 2008*)), and either move or become inaccessible, in many cases before the papers have been published. This can happen either because the web hosting is not stable, or fails, or because the domain providing the URI is not maintained. To alleviate these difficulties with MITAP, neither the URIs nor the hosting of the documents require active maintenance by the authors of this paper. We have used a permanent identifier (http://w3id.org/ontolink/mitap) thereby providing a re-direction step. The primary hosting of the resource is provided by Internet Archive (http://archive.org). Both of these organisations have sustainability as their key aim. Maintenance of MITAP documents as live, web-resources therefore requires no active contribution from the authors of this paper. Either archive.org or w3id.org need to fail before MITAP cannot be maintained at the stated location.

We have also given explicit consideration to data formats for our documents. Resources are available as Word docs and PDFs, providing typographical elegance and familiarity to the community, but also an extremely simple HTML representation, ensuring vendor-neutrality and future-proofing of the knowledge contained within them.

## DISCUSSION

The tolAPC community has advanced significantly in the last 15 years, from the initial discoveries of how to culture APC with potent immune-regulatory properties, through to clinical trials. At the current time, however, most investigators are operating as a "cottage industry": individual labs working with different cell isolation techniques, procedures to induce stable tolerogenic properties in the cells, and investigating different therapeutic outcomes. Against this heterogeneous background, it becomes difficult to compare investigations between different laboratories, raising concerns about reproducibility (*Ioannidis, 2005*; *Boulton, 2016*).

The MITAP document that we have described here will, of course, not address all of these issues, however as a minimum guidelines tool for reporting of protocols, it should at least address the issue of data heterogeneity; different laboratories may still use different

protocols to generate tolAPC, but at least we will be clear about this diversity, and have a greater understanding of what these differences are.

We have been careful in the process of building MITAP not to produce a document that is overly complex. Unlike many other minimum information documents, we have actively tested it with independent, essentially untrained annotators (*test subjects*). We have shown that it is possible, under these circumstances, to provide the information required in a clear and consistent manner. We believe that it should be possible for authors to complete the MITAP document in approximately 30 min. We also believe that this process should not be a burden to researchers; in our experience, we have adopted use of the MITAP document in writing our own papers, and believe that having a clear checklist to follow simplifies rather than confuses.

The process of building the MITAP document has been a highly collaborative one which has been a useful exercise in and of its own right: the degree of heterogeneity in the community has come as a surprise to many of those involved in this work. By analyzing extant papers, we have also shown that there is a discrepancy between the information that the community considers important and wants to receive about a tolAPC production protocol and the information that they actually report. For example, very few papers describe the viability of tolAPC at the end of the production process, although we consider it extremely likely that most researchers do actually measure this. Moreover, we can see no *a priori* reason why culture media largely is reported while culture container largely is not.

We believe that wide adoption could also contribute toward greater consideration of data management in the tolerogenic cell therapies community in general. For example, the MIATA project was recently extended to include NK cells (called MIANKA), and a similar approach could be taken by MITAP to include other regulatory cells, such as myeloid-derived suppressor cells, mesenchymal stem cells, and even T regulatory cells. Furthermore, unlike many minimum information efforts, we have considered upfront long-term sustainability of the documents and the URIs used to refer to them. Our sustainability plan is robust and does not require active involvement of any of the authors.

Of course, there is much work remaining. The production of tolAPC is only a small part of the overall process. Moreover, we are only attempting to standardize the reporting of protocols, and not the standardisation of the protocols themselves. Finally, we currently have no common location or database which can store the experimental data, such as exists for instance with microarray or genomic data. MITAP is a first step, laying the groundwork for further collaboration.

## ACKNOWLEDGEMENTS

We would like to thank the volunteers (*test subjects*) who participated in testing MITAP. We also thank Prof Ranjeny Thomas for providing feedback on the manuscript and MITAP document.

### Funding

This work was supported by a grant from the European Cooperation in Science and Technology (COST) for the AFACTT project (Action to Focus and Accelerate Cell-based Tolerance-inducing Therapies; BM1305). COST is part of the EU Framework Programme Horizon 2020. The funders had no role in study design, data collection and analysis, decision to publish, or preparation of the manuscript.

### Grant Disclosures

The following grant information was disclosed by the authors:
The European Cooperation in Science and Technology (COST): BM1305.

### Competing Interests

The authors declare there are no competing interests.

### Author Contributions

- Phillip Lord and Catharien M.U. Hilkens conceived and designed the experiments, performed the experiments, analyzed the data, wrote the paper, reviewed drafts of the paper, drafted the MITAP document.
- Rachel Spiering and Eva M. Martinez-Caceres performed the experiments, analyzed the data, prepared figures and/or tables, reviewed drafts of the paper, drafted the MITAP checklist, provided feedback on the MITAP document.
- Juan C. Aguillon, Amy E. Anderson, Silke Appel, Daniel Benitez-Ribas, Anja ten Brinke, Femke Broere, Nathalie Cools, Maria Cristina Cuturi, Julie Diboll, Edward K. Geissler, Nick Giannoukakis, Silvia Gregori, S. Marieke van Ham, Rachel A. Harry, James A. Hutchinson, John D. Isaacs, Irma Joosten, Cees van Kooten, Ascension Lopez Diaz de Cerio, Tatjana Nikolic, Haluk Barbaros Oral, Ljiljana Sofronic-Milosavljevic, Thomas Ritter, Paloma Riquelme, Angus W. Thomson, Massimo Trucco and Marta Vives-Pi reviewed drafts of the paper, provided feedback on the MITAP document and checklist.
- Staci Lattimer and Lindsay F/ Marshall contributed reagents/materials/analysis tools, reviewed drafts of the paper, provided feedback on the MITAP document and checklist.

### Data Availability

The raw data has been supplied as Supplementary File.

### Supplemental Information

Supplemental information for this article can be found online at http://dx.doi.org/10.7717/peerj.2300#supplemental-information.

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
