# Peer review of "Minimum information about tolerogenic antigen-presenting cells (MITAP): a first step towards reproducibility and standardisation of cellular therapies"

_PeerJ, doi:10.7717/peerj.2300_

## Round 0.1 · accepted · Accept

The paper was deemed important and laying the groundwork to conduct and interpret cell based studies in this area.

·

Basic reporting

No comments

Experimental design

The authors set up a standardized reporting method, and then test concordance on terms and prior literature for level of information corresponding to this new reporting method.

Validity of the findings

No Comments

Additional comments

Setting up these standards for reporting methods for producing tolerogenic APCs will improve future efforts in this area. The checklist and MITAP document are clear and should be easy to follow and implement when reporting future studies.

Reviewer 2 ·

Basic reporting

This is a report by a group of investigators who are committed to the study of tolerogenic Antigen-Presenting-Cells (tolAPCs). This is a timely paper as early studies indicate their potential value in the treatment of autoimmune diseases and the prevention of transplant rejection. This group decided to get together to generate a minimum information model for tolAPC called MITAP with the goal of providing a reporting framework that will allow a comparison of the various batches of tolAPCS used in the different studies.

Experimental design

There is no primary experimental research in this manuscript. But the authors have brainstormed to come up with a set of recommendations to allow a more accurate reporting of experiments using tolAPCs. The recommendations are well choosen allowing an accurate description of the cellular preparation and experimental protocol without imposing the use of a specific platform.

Validity of the findings

The intent is of great value as this initiative will permit the community to better assess the data generated with tolAPCs. This is particularly important as these studies will in many cases be performed in patients at high costs. Thus, a detailed and accurate report of the overall trial is a welcome endeavor. The manuscript is well written. The aspects that are considered and discussed are appropriate. The authors did not fall into the trap that a single platform should be used. Rather they propose a framework that will permit a detailed comparison of the used platforms. Hopefully, it will permit them to identify which platform is the most efficinet to induce tolerance mechanisms.

Additional comments

I strongly support the publication of this manuscript in PeerJ or elsewhere.